# Co-Expression of CD34, CD90, OV-6 and Cell-Surface Vimentin Defines Cancer Stem Cells of Hepatoblastoma, Which Are Affected by Hsp90 Inhibitor 17-AAG

**DOI:** 10.3390/cells10102598

**Published:** 2021-09-29

**Authors:** Mieun Lee-Theilen, Julia R. Hadhoud, Giulietta Volante, Delaine D. Fadini, Julia Eichhorn, Udo Rolle, Henning C. Fiegel

**Affiliations:** 1Department of Pediatric Surgery and Pediatric Urology, University Hospital, Goethe University Frankfurt, 60590 Frankfurt, Germany; s8061954@stud.uni-frankfurt.de (J.R.H.); Delaine.Fadini@gmx.de (D.D.F.); juleich@yahoo.de (J.E.); Udo.Rolle@kgu.de (U.R.); Henning.Fiegel@kgu.de (H.C.F.); 2Department of General, Visceral and Transplant Surgery, University Hospital, Goethe University Frankfurt, 60590 Frankfurt, Germany; Giulietta.Volante@kgu.de

**Keywords:** cancer stem cells, hepatoblastoma, CSC marker, Hsp90 inhibitor, 17-AAG

## Abstract

Cancer stem cells (CSCs) are nowadays one of the major focuses in tumor research since this subpopulation was revealed to be a great obstacle for successful treatment. The identification of CSCs in pediatric solid tumors harbors major challenges because of the immature character of these tumors. Here, we present CD34, CD90, OV-6 and cell-surface vimentin (csVimentin) as reliable markers to identify CSCs in hepatoblastoma cell lines. We were able to identify CSC characteristics for the subset of CD34^+^CD90^+^OV-6^+^csVimentin^+^-co-expressing cells, such as pluripotency, self-renewal, increased expression of EMT markers and migration. Treatment with Cisplatin as the standard chemotherapeutic drug in hepatoblastoma therapy further revealed the chemo-resistance of this subset, which is a main characteristic of CSCs. When we treated the cells with the Hsp90 inhibitor 17-AAG, we observed a significant reduction in the CSC subset. With our study, we identified CSCs of hepatoblastoma using CD34, CD90, OV-6 and csVimentin. This set of markers could be helpful to estimate the success of novel therapeutic approaches, as resistant CSCs are responsible for tumor relapses.

## 1. Introduction

Hepatoblastoma is the most common pediatric liver tumor and is usually diagnosed within the first 5 years of life [1]. It is now well accepted that hepatoblastomas originate from immature and pluripotent cells, which became tumorigenic during the embryonal and fetal development of the liver. The International Pediatric Liver Consensus Classification subdivides the tumors by histology in fetal, embryonal, cholangioblastic, macrotrabecular and small-cell undifferentiated (SCUD) patterns. However, these tumors are of high heterogeneity with often closely intermixed histological components [2,3,4]. Some tumors, such as the SCUD subtype, show poor responses to chemotherapy [1,5].

CSCs are a minor fraction of cancer cells and, like normal stem cells, possess the ability to self-renew, but also provide multiple mature progenies [6]. In general, this fraction of the tumor cells was found to be highly tumorigenic and responsible for drug resistance, metastasis and high relapse rates in certain cancers [6]. CSCs have been extensively investigated in adult tumors, as they can be more easily distinguished from the rest of the tumor cells, which have a usually more differentiated phenotype. In contrast, identifying CSCs within pediatric tumors, such as hepatoblastoma, represents a great challenge, as they share similar immature features with the vast majority of the tumor cells. Well-known stem-cell-specific proteins are therefore not solely expressed by CSCs but also to a high level in other tumor cells. In addition, tumors originating from the liver with its regenerating capacity, even in the adult age, definitely adds another difficulty to the search for reliable markers, as it can also be seen in the research for CSC of hepatocellular carcinoma [7,8,9].

A few studies have previously attempted to identify CSC markers in hepatoblastoma [10,11,12]. One of them, namely, the well-studied CD133, was investigated for its role in the migration behavior of hepatoblastoma cells [13]. Further studies analyzed the expression of CD90 and CD34 in hepatoblastoma specimens using immunohistochemistry and could observe a specific expression of these markers on suggested stem-like cells [4,14]. The “oval cell” antibody OV-6 was also used to detect suspected cancer stem cells in hepatoblastoma tissue specimens [2,15,16]. Since the presence of CSCs remains a major problem when treating cancer in general, a reliable set of CSC markers for hepatoblastoma could contribute to a better treatment outcome for this tumor. It would provide researchers with a useful tool to control novel treatment approaches for the successful elimination of CSCs. 

In our study, we aimed to investigate the suggested hepatoblastoma CSC markers CD90, CD34 and the OV-6 antibody in two hepatoblastoma cell lines (HuH6 and HepG2). Additionally, we sought to investigate vimentin on the cell surface as a potential fourth marker of hepatoblastoma CSCs [17]. In contrast to former studies, which only proposed CSC profiles, our purpose was to investigate the cells for CSC characteristics in more detail by performing several assays. Finally, we sought to find a hepatoblastoma-CSC-specific drug. 

## 2. Materials and Methods

### 2.1. Cell Culture

The HepG2 cell line was kindly provided by Elsie Oppermann of the General Surgery Department, University Hospital Frankfurt am Main. The HuH6 cell line was kindly provided by Prof. Kappler of the Pediatric Surgery Department, University Hospital Munich. HepG2 and HuH6 cells were grown in Dulbecco’s Modified Eagle Medium + Glutamax^TM^-I (Gibco, Carlsbad, CA, USA), supplemented with 10% fetal bovine serum (FBS, Gibco, Carlsbad, CA, USA) and 2.5 mg/mL Gentamicin (Gibco, Carlsbad, CA, USA) at 37 °C, 5% CO_2_ and 90% humidity. The cells were passaged via treatment with Accutase (Sigma-Aldrich, St. Louis, MI, USA) for 10 min at RT. For the treatment with Cisplatin (TEVA GmbH, Ulm, Germany), 1.5 × 10^6^ HuH6 cells or 4 × 10^6^ HepG2 cells were plated out the day before in 75 cm^2^ tissue culture flasks. The next day, the cells were treated with Cisplatin at different concentrations for 72 h and subjected to further analyses. For the treatment with 17-AAG (17-N-allylamino-17-demetoxygeldanamycin; Tocris by Biotechne, Minneapolis, MN, USA), the cells were plated out the same way. The treatment with 17-AAG was performed for 48 h. Magnetic separation of CD34, CD90 or OV6-positive cells was performed according to the respective protocols provided by Miltenyi Biotec (Bergisch Gladbach, Germany).

### 2.2. MTT (3-(4,5-dimethylthiazol-2-yl)-2,5-diphenyltetrazolium bromide) Assay

The cells were plated out with a density of 5000 cells per well in a 96-well plate. The next day, the cells were treated with the respective drugs in triplicates. After the incubation time, the MTT assay was performed according to the manufacturer’s protocol (Cell Proliferation Kit, Roche Diagnostics GmbH, Penzberg, Germany).

### 2.3. Tumorsphere Assay

The cells were incubated in serum-free DMEM-Glutamax^TM^-I with 1 × B27 supplement (Fisher Scientific GmbH, Schwerte, Germany), 1 × N2 supplement (Fisher Scientific GmbH, Schwerte, Germany), 20 ng/mL EGF (Sigma-Aldrich, St. Louis, MI, USA) and 20 ng/mL FGF (Peprotech, Rocky Hill, CT, USA) on low-attachment 24-well plates (Greiner Bio-One International GmbH, Frickenhausen, Germany). The growing spheres were incubated for 7 days and passaged by separating the spheroid cells with Accutase and replating in fresh tumorsphere media on new plates.

### 2.4. Flow Cytometry

The cells were stained for the surface expression of CD34 using a BV421-conjugated anti-CD34 antibody (clone 581, BD Horizon, Franklin Lakes, NJ, USA), the expression of CD90 using an FITC-conjugated anti-CD90 antibody (clone 5E10, BioLegend, Koblenz, Germany), the expression of vimentin using a PE-conjugated anti-vimentin antibody (clone D21H3, Cell Signaling Technology, Frankfurt, Germany) and the expression of CXCR4 using a PE-conjugated anti-CXCR4 antibody (clone12G5, R&D Systems, Biotechne, Wiesbaden, Germany). The cells were also stained with an APC-conjugated OV-6 antibody (R&D Systems, Biotechne, Wiesbaden, Germany), Data were acquired on a BD FACSCanto II (BD Biosciences, San Jose, CA, USA) and analyzed using the Flowjo software v7.2.5 (BD Life Sciences, Ashland, OR, USA).

### 2.5. Migration Assay

HuH6 and HepG2 cells (5 × 10^5^ in 2 mL of growth medium without FBS) were placed in a membrane insert with 8 µm pores (Corning, Inc., Amsterdam, Netherlands) and the inserts were placed into the wells of a 6-well plate (Sarstedt, Nuernbrecht, Germany), which were coated with 2 mL growth medium with 10% FBS. The cells were incubated at 37 °C for 24 h. After incubation, the migrated cells at the bottom side of the insert membrane and the non-migrated cells in the upper side of the insert membrane were scraped off and RNA was extracted, cDNA synthesized and subjected to qPCR. For cell counting, the non-migrated cells were wiped off with cotton swabs and the migrated cells on the bottom side of the insert membranes were fixed with 2% Glutardialdehyde (Merck, Darmstadt, Germany), stained with Mayer’s hematoxylin (Applichem GmbH, Darmstadt, Germany) and the cell numbers were counted per 0.25 mm^2^ under an inverted microscope.

### 2.6. RNA Extraction and Transcription Analysis Using Real-Time PCR

RNA was extracted from the cells using the Extractme Total RNA Kit (blirt S.A., Gdansk, Poland) according to the manufacturer’s protocol. cDNA was generated using the iScript cDNA Synthesis Kit (Bio-Rad Laboratories, Inc., Hercules, CA, USA). Real-time PCR was performed using iTaq Universal SYBR Green Supermix (Bio-Rad Laboratories, Inc., Hercules, CA, USA) with a Stratagene Mx3005P machine (Agilent, Santa Clara, CA, USA). qPCR data were calculated based on the mean of two experimental replicates and all qPCR experiments were repeated at least 3 times. Dissociation curves were controlled to confirm the amplification of only one PCR product. All quantifications were normalized to an endogenous ACTB control and calculated using the 2^(−^^ΔΔCt)^ method. All the primers used spanned introns and are listed in Appendix A. Primers were designed with Primer3Plus or retrieved from publications.

### 2.7. Statistical Analysis

The results are presented in diagrams as means and standard deviations. A non-parametric two-tailed Wilcoxon signed-rank test was used for comparisons between two groups and a non-parametric Dunn’s multiple comparisons test was used for comparisons between several groups. The *p*-values < 0.05 were considered statistically significant.

## 3. Results

### 3.1. Cancer Stem Cell Markers CD34, CD90, OV-6 and Cell-Surface Vimentin Were Co-Expressed on a Subset of Hepatoblastoma Cells

In order to define a set of markers for the identification of CSCs in hepatoblastoma, we analyzed the expression of CD133, CXCR4, CD34 and CD90 in the hepatoblastoma cell lines HepG2 and HuH6 using flow cytometry. In addition, we also tested the “oval cell” antibody OV-6. 

FACS analyses revealed that CD133 was expressed by almost 100% of the HuH6 cells and about 60% by the HepG2 cells (data not shown). We concluded that CD133 might be a general marker for immature cells, but can be excluded from our list of possible markers for hepatoblastoma CSCs. We then analyzed the expression rate of CXCR4, CD34, CD90 and binding of the OV-6 antibody. Figure 1A shows that 13.8% of HepG2 cells were positive for CXCR4, 10% positive for CD90, 9.9% positive for CD34 and 10.1% positive for OV-6. The HuH6 cells revealed higher numbers with 39.5% for CXCR4, 15.1% for CD90, 15.4% for CD34 and 17.4% for OV-6 (Figure 1B). 

We also investigated the expression of vimentin on the cell surface. According to previous studies, vimentin can be detected outside of the cell. Therefore, we utilized an antibody that was raised against the N-terminal part of the protein, which can be found outside the cell [18]. Since the cells were not fixed and permeabilized, the binding of the antibody only occurred on the cell surface. The results revealed that cell-surface vimentin (csVimentin) could be detected on 9.1% of HepG2 cells and 15% of HuH6 cells. The results showed that in HepG2 and HuH6 cells, the detection rates of CD34, CD90, OV-6 and csVimentin were comparable in each cell line, except for CXCR4 (Figure 1A,B).

The detection of these cell-surface proteins at a similar expression level led us to speculate that the surface proteins might be expressed together on a certain subset of cells. To verify this, we stained HepG2 (Figure 1C) and HuH6 (Figure 1D) cells simultaneously for CD34, OV-6 and CD90, as well as CD34, OV-6 and csVimentin, respectively. We observed, that CD34/OV-6 double-positive cells were also positive for CD90 and csVimentin. Overall, the vast majority of the cells were either CD34^−^OV-6^−^CD90^−^csVimentin^−^ or CD34^+^OV-6^+^CD90^+^csVimentin^+^. Based on this observation, we assumed that the percentage of any of these markers could be referred to as the percentage of the other markers.

### 3.2. CD34^+^CD90^+^OV-6^+^csVimentin^+^ Subset Revealed Pluripotency Features and Increased Expression of EMT Markers

Since a smaller percentage of the cells was positive for the four markers (~10% in HepG2 cells and ~15% in HuH6 cells), we investigated this subset for CSC features and analyzed the mRNA levels of pluripotency markers and EMT (endothelial-to-mesenchymal transition) markers using qPCR. 

First, we enriched the population for each of CD34, CD90 or OV-6 by using the MACS technique and measured the expression of the other markers using FACS. As expected, when the cells were sorted for CD90, the enriched CD90 fraction showed an increased expression of CD34 and csVimentin and OV-6 binding compared to the depleted population (Appendix A) and the same was true for the other sortings (CD34-MACS or OV-6-antigen-MACS, data not shown). The total RNA of the enriched and the depleted fractions were extracted and subjected to qPCR analyses.

The qPCR results shown in Figure 2 revealed that the pluripotency markers Oct4 and Nanog were significantly increased in the CD34-enriched populations of HepG2 (CD34^+^ in Figure 2A) and HuH6 cells (CD34^+^ in Figure 2C). The proto-oncogene c-myc showed a minor but significant increase in the CD34 enriched populations as well. Epithelial cell adhesion molecule (EpCAM), which is expressed in premature liver cells, alpha-fetoprotein (AFP), which is the biomarker for hepatoblastoma, and albumin, which is considered as a marker for mature hepatocytes, did not show different expressions in the enriched and depleted fractions (Figure 2A,B). The same results were also observed when the cells were enriched for CD90 (Appendix A). 

Additionally, we measured the expression of factors involved in the EMT. The CD34-enriched fraction showed a significantly increased expression of the EMT transcription factors SNAI1 and TWIST1 and the mesenchymal marker vimentin, but a decreased expression of the endothelial markers E-cadherin and occludin in both cell lines, as shown in Figure 2B,D. Again, the results were confirmed when the cells were enriched for CD90 (Appendix A).

### 3.3. CD34^+^CD90^+^OV-6^+^csVimentin^+^ Cells Showed Self-Renewal Ability and Increased Migration Behavior

Tumorsphere assays were also used to confirm the stem cell property of the CD34^+^OV-6^+^CD90^+^csVimentin^+^ subpopulation, as this assay was based on the ability of self-renewal. Cells were incubated in FBS-free media on low-attachment plates with growth factors (FGF, EGF). We cultivated HepG2 cells under these conditions and, over time, we could observe the formation of spheres. We passaged the spheres three times in a weekly period by separating the spheroid cells and incubating them in fresh media on new plates (Figure 3A). The total RNA of half of the cells was harvested and reverse transcribed into cDNA and subjected to qPCR analyses and the expression of CD34, CD90, KRT14 (one of the antigens of OV-6 antibody), Oct4, Nanog, c-myc and albumin was measured (Figure 3B). We observed increased, although non-significant, expressions with every passage number for CD34, CD90, KRT14, Oct4 and Nanog, whereas the expressions of c-myc and albumin remained unchanged. This gave us evidence that all four surface markers, along with the pluripotency markers Oct4 and Nanog, were expressed in a self-renewing subpopulation.

As it was speculated that CSCs are the driving force of metastasis, we further performed migration assays with Boyden chambers using the HuH6 and HepG2 cells. After 24 h, we analyzed the migrated (m) and the non-migrated (nm) fractions using qPCR for their gene expressions of CD34, CD90, KRT14, Oct4, Nanog, SNAI1, Twist1, vimentin, E-cadherin and occludin (Figure 3C–F). Almost all factors were significantly increased in the migrated cell fraction compared to the non-migrated fraction, revealing that CD34^+^CD90^+^OV-6^+^csVimentin^+^ cells predominantly migrated, whereas E-cadherin and occludin showed partially significantly decreased values. This was confirmed by performing the assay with CD90-depleted and CD90-enriched cells (Appendix A). We observed an increased percentage of cells that migrated in the CD90-enriched fraction.

The tumorsphere assays revealed that the CD34^+^CD90^+^OV-6^+^csVimentin^+^cells had the ability of self-renewal, which is one of the major features of cancer stem cells. In addition, the subset showed an increased migration behavior, which could reflect the role of cancer stem cells in metastasis.

### 3.4. Cisplatin Treatment Caused Positive Selection of CD34^+^CD90^+^OV-6^+^csVimentin^+^ Cells

Cancer stem cells are responsible for resistance toward chemotherapy. Treatment with a chemotherapeutic drug could therefore result in an enrichment of the CD34^+^CD90^+^OV-6^+^csVimentin^+^ subset. We treated HuH6 and HepG2 cells with Cisplatin (1, 2.5, 5 and 7.5 µg/mL), which is the gold standard for hepatoblastoma treatment.

The results of the MTT assays showed that at the maximal concentration of 7.5 µg/mL, the relative number of viable cells went significantly down to 16.5% for HepG2 cells and to 5.8% for HuH6 cells compared to untreated cells, showing that the vast majority did not survive the treatment, but a small number of cells was resistant (Figure 4A,B). We analyzed the cells using FACS and could see that the Cisplatin treatment resulted in significantly increasing the percentages of CD34^+^ OV-6^+^ cells, with the highest percentages for HepG2 cells at 5 µg/mL with 57.9% and for HuH6 cells at 7.5 µg/mL with 62.6%, as shown in Figure 4C,D (Appendix A). Figure 1C,D shows that the expressions of CD90 and csVimentin were almost identical with CD34 and OV-6 such that we could refer to the percentage of the CD34^+^OV-6^+^ cells as the percentage of the positive cells for all four markers. The values of CD34^+^OV-6^+^ cells that are presented here are therefore the number of CD34^+^CD90^+^OV-6^+^csVimentin^+^ cells.

Finally, we isolated the RNA of the surviving cells and performed qPCR in order to measure the transcriptions of Oct4, Nanog, SNAI1, Twist1, vimentin, albumin and E-cadherin (Figure 4E–R). The transcript levels of Oct4, Nanog, SNAI1, Twist1 and vimentin were increased in HepG2 cells (Figure 4E–I) and in HuH6 cells (Figure 4L–P) to more than fivefold. Statistical analyses revealed significant differences, which are indicated in the diagrams.

Taken together, treatment with Cisplatin resulted in a positive selection of the CD34^+^CD90^+^OV-6^+^csVimentin^+^ subset, confirming chemo-resistant behavior.

### 3.5. 17-AAG Affected the CD34^+^CD90^+^OV-6^+^csVimentin^+^ Subset

17-AAG, which is an inhibitor of the chaperone Hsp90, depicts a possible strategy to target cancer stem cells. In order to investigate whether Hsp90 inhibition can affect the CD34^+^CD90^+^OV-6^+^csVimentin^+^ subset, we treated HuH6 and HepG2 cells with 17-AAG for 48 h with concentrations of 0.1 and 0.25 µM; the MTT assays revealed that the overall viability of about 60% (at a concentration of 0.25 µM) was not as decreased as observed with Cisplatin in HuH6 cells, indicating that 17-AAG did not kill the cells to the same extent (Figure 5A). A statistical analysis also revealed that the decrease was not significant. When we analyzed the numbers of the CD34^+^ OV-6^+^ cells using flow cytometry, the percentage was lowered significantly with 0.1 µM 17-AAG, but not with 0.25 µM 17-AAG, in HuH6 cells after 48 h (Figure 5B and Appendix A). Again, we could assume that this percentage stood for the CD34^+^CD90^+^OV-6^+^csVimentin^+^ subset. In contrast, the percentage of the positive cells increased substantially in HepG2 cells at a concentration of 0.25 µM (Appendix A), whereas the viability decreased to the same extent as when treated with Cisplatin (Appendix A). Therefore, we discontinued our experiments with HepG2 cells at this point. We performed qPCR analyses with the HuH6 cells and observed that the 17-AAG-treated cells had partially significantly decreased Oct4, Nanog, CD34, CD90, KRT14, SNAI1 and Twist1 mRNA expressions compared to the control cells, whereas the EpCAM, albumin, c-myc, vimentin and E-cadherin mRNA levels were not affected by the treatment (Figure 5C–N). This indicated that the treated cells had an altered expression profile with less pluripotency. Next, we performed tumorsphere assays with the HuH6 cells and added 0.1 µM 17-AAG in order to investigate whether 17-AAG affected the formation of the spheres and therefore inhibited their capacity for self-renewal (Figure 5O,P). Since the concentration of 0.1 µM revealed significantly decreased percentages of CD34^+^OV-6^+^ cells, we decided to use the lower concentration. Eleven days after incubation, the spheres of the cells treated with 17-AAG were significantly smaller than those of the untreated cells (Figure 5Q). The spheres also appeared less stable. Thus, the observations showed that 17-AAG indeed affected the CSC expansion, either by killing or modifying the cells.

We then analyzed the effect of 17-AAG in combination with Cisplatin, which caused a positive selection of the CD34^+^CD90^+^OV-6^+^csVimentin^+^ cells (Figure 4B). Therefore, HuH6 cells were pre-treated with 0.1 µM 17-AAG for 48 h and afterward with 2 µg/mL Cisplatin for another 72 h and compared to cells that were only treated with 2 µg/mL Cisplatin for 72 h or left untreated. The MTT assays clearly showed that treatment with 17-AAG did not further decrease the viability of the Cisplatin-treated cells (Figure 6A).

However, when we performed flow cytometry, the percentage of CD34^+^ OV-6^+^ cells (and, therefore, of all four proteins of the marker positive subset) of the double-treated cells with ~15% was not as high as the percentage of cells that were only treated with Cisplatin (~30%). The result was comparable to that of the untreated fraction (Figure 6B and Appendix A). 

Finally, we measured the expressions of Oct4, CD34, CD90, vimentin and EpCAM using qPCR and could observe that the double-treated cells did not show any elevated amounts for Oct4, CD34 and CD90 of the Cisplatin mono-treated cells that we observed in our previous experiments (Figure 4L–P). The expression levels were again comparable to those of the control cells (Figure 6C–G).

Combining the MTT data with the FACS and qPCR data revealed that 17-AAG did not necessarily kill the CSCs but rather changed the expression profile. 

## 4. Discussion

The lack of markers for hepatoblastoma CSCs complicates the search for successful therapy strategies against this tumor entity. This led us to investigate widely discussed markers for their usefulness. Our results showed that the candidates CD133, CXCR4, CD34, CD90 and OV-6 binding were detected at different expression levels in the two investigated hepatoblastoma cell lines (HuH6 and HepG2). Since CD133 was expressed on HuH6 and HepG2 cells almost ubiquitously, we removed it from our list for hepatoblastoma CSCs. The expression rate of almost 40% for CXCR4 on HuH6 cells also questioned its role as a CSC marker, even though its role in stemness and drug resistance, as well as metastasis in other cancers, was reported [19]. Altogether, the high expression levels of CD133 and CXCR4 could be explained by the embryonal nature of the tumor. In contrast, we detected CD34, CD90 and csVimentin and the binding of OV-6 antibody only on a minority of cells. Co-staining revealed that all four markers were expressed on the same subset of cells. The choice of CD34 and CD90, which are primarily known as hematopoietic stem cell markers, was based on the fact that the embryonal liver in the early stage is the initial source of all hematopoietic cells. Since the early immature liver and hematopoietic cells are generated in the same organ, hepatoblastoma CSCs might share a similar profile with hematopoietic stem cells. Furthermore, we decided to use the OV-6 antibody, as other studies already detected OV-6-positive cancer stem cells in HCC [20]. csVimentin was chosen as a potential hepatoblastoma CSC marker since csVimentin was found on cancer stem cells of other tumor entities [17,18,21,22,23]. Additionally, being the major cytoskeletal component of mesenchymal cells, vimentin is considered as a marker for cells that have undergone the EMT process. This embryonic program is utilized by tumors to generate cells with invasive properties via the loss of epithelial differentiation and gain of a mesenchymal phenotype. Using this transformation, cancer cells are able to leave the primary tumor site and travel to form metastases. CSCs are said to be responsible for this process. By investigating their pluripotency ability, we noticed that the CD34^+^CD90^+^OV-6^+^csVimentin^+^ cells had increased mRNA levels of the transcription factors Oct4 and Nanog, which are two essential pluripotency factors and are highly expressed in embryonic stem cells, as well as CSCs [24,25,26]. SNAI1, Twist1 and vimentin were found to be increased as well. SNAI1 and Twist1 have a driving role in the EMT process [27]. By performing tumorsphere assays, which is an established method to selectively culture CSCs in vitro, we observed that the proliferating sphere cells had increased CD34 and CD90 expressions and an OV-6 detection rate compared to normal cells. The results of the migration assays revealed an increased migration behavior. When we treated both cell lines with Cisplatin, we saw that the resistant cells were revealed to be mainly the subset of CD34^+^OV-6^+^CD90^+^csVimentin^+^ cells. This finding confirmed a chemo-resistant behavior, which is one of the main characteristics of CSCs [28].

Our Cisplatin treatment results coincided with previous findings of other studies that showed that Cisplatin is not able to kill CSCs but rather causes a positive selection [28,29]. This explains why a treatment based on only cytostatic modes does not always achieve a sufficient response and results in tumor relapses. So far, hepatoblastoma therapy does not consider a targeted treatment against CSCs. One approach to specifically target CSC represents the inhibition of the Hsp90 complex [30,31]. Hsp90 is a chaperone protein that is responsible for the stabilization of many proteins, such as crucial oncogenes, and thus was revealed to have an oncogenic function by itself [32]. By inhibiting Hsp90, stemness factors could be destabilized and the consequence might be cell death or the loss of stemness. Cancer cells usually have Hsp90 in a complex and Hsp90 inhibitors primarily target Hsp90 in the complex with other chaperones [33]. We tested the Hsp90 inhibitor 17-AAG (17-N-allylamino-17-demethoxygeldanamycin), also known as Tanespimycin, which is an analog of the antibiotic Geldanamycin for our purposes [34]; we observed a significant decrease in the CSC population in HuH6 cells when treated with 17-AAG. However, when we analyzed the viability, we could observe that the general cytotoxic effect of 17-AAG was not very pronounced. We speculated that the decrease in the CSC population was not based on triggered cell death due to 17-AAG, but rather a change in the differentiation status. This was suggested by the decreased expression of the pluripotency markers Oct4 and Nanog of the treated cells. Furthermore, tumorsphere assays revealed that treatment with 17-AAG disrupted the formation of the spheres and, therefore, the self-renewal capability of the cells. The decrease in CSC numbers due to the 17-AAG treatment, however, was not observed in HepG2 cells. Instead, 17-AAG showed the same strong cytotoxic effect as Cisplatin, triggering cell death in preferentially normal cancer cells. A possible explanation could be that in HepG2 CSCs, the key regulators that were responsible for cancer cell stemness were not stabilized by Hsp90 so that inhibition of Hsp90 did not affect these proteins and the stemness was not disturbed. Instead, it seems that 17-AAG induced the decrease of cell death inhibiting factors which resulted in cell death. It would still be of great interest to investigate the impact of other Hsp90 inhibitors on HepG2 CSCs, which have different mechanisms that affect the function of Hsp90. 

Since treatment with Cisplatin resulted in a positive selection of the CSC subset, the question we also addressed was a possible effect of 17-AAG on HuH6 cells when given before Cisplatin. The results revealed that 17-AAG given beforehand indeed diminished the number of CSCs since the expected Cisplatin-mediated increase of CSCs was not as pronounced as in the Cisplatin mono-treated group. The accompanying viability assays indicated that the cells were instead forced toward a more differentiated status, as the viability of the double-treated cells was comparable to that of the Cisplatin mono-treated cells. The qPCR results supported this hypothesis by revealing decreased mRNA levels of the pluripotency markers Oct4 and Nanog for the double-treated cells. Our results are supported by a recent study in which 17-AAG inhibited stem-cell-like properties in osteosarcoma cells via the Hedgehog signaling pathway [35]. Thus, a preceding administration of 17-AAG could terminate the stemness of the cells by preventing Hsp90 from stabilizing pluripotency factors. The loss of stemness would assimilate them to the rest of the cancer cells, making them equally susceptible to a subsequent Cisplatin treatment. The identification of the responsible stemness factors would help to clarify the nature of CSCs and their dependences. A great advantage of Hsp90 inhibitors is the lack of a direct influence on the integrity of the genome by, e.g., by causing direct DNA damage. This is important since in early childhood, organic and cellular development are ongoing and DNA damage as part of excessive cytostatic therapies can cause secondary malignancies. Overall, the use of 17-AAG as part of a mono- or multimodal chemotherapy regimen is promising, but still to be investigated. So far, a phase I study was conducted for children with refractory tumors in order to investigate the tolerated dosage and side effects [36]. 

## 5. Conclusions

We presented CD34, CD90, OV-6 and csVimentin together as a reliable combination of markers to identify hepatoblastoma CSCs. By using various assays, we were able to identify CSC characteristics, such as pluripotency, self-renewal, EMT, migration and chemo-resistance for this subset. In addition, and for the first time, we showed that the Hsp90 inhibitor, namely, 17-AAG, might be a suitable drug to selectively target hepatoblastoma CSCs. Further studies will be needed to fully understand the effect of 17-AAG on CSCs in hepatoblastoma. 

## Figures and Tables

**Figure 1 cells-10-02598-f001:**
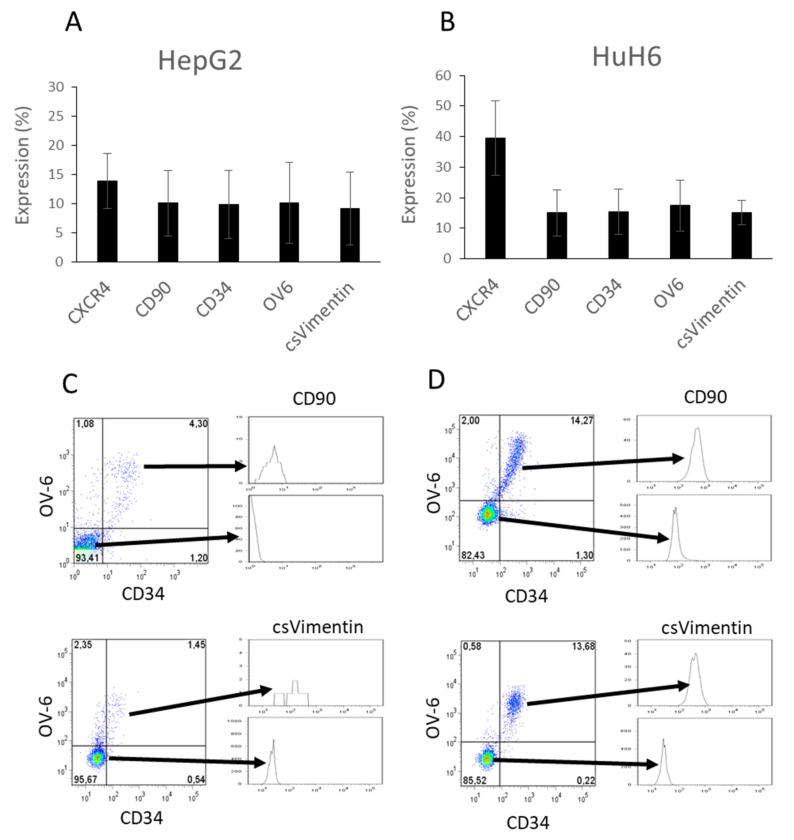
CD34, CD90 and csVimentin expressions and OV-6 binding were detected on a subpopulation of HepG2 and HuH6 cells. (**A**) The expressions of CXCR4, CD90, CD34 and vimentin and the binding of OV-6 on HepG2 cells and (**B**) on HuH6 cells were measured using flow cytometry. The box plots show the mean of at least 10 independent measurements with error bars depicting the standard deviation from the mean. (**C**) HepG2 and (**D**) HuH6 cells were simultaneously stained with α-CD34, OV-6 and α-CD90 antibodies and with α-CD34, OV-6 and α-vimentin antibodies, respectively. The cells were gated for CD34 and OV-6 double-positive and CD34 and OV-6 double-negative cells and subsequently analyzed for CD90 or csVimentin expression. These are representative results of at least 10 experiments.

**Figure 2 cells-10-02598-f002:**
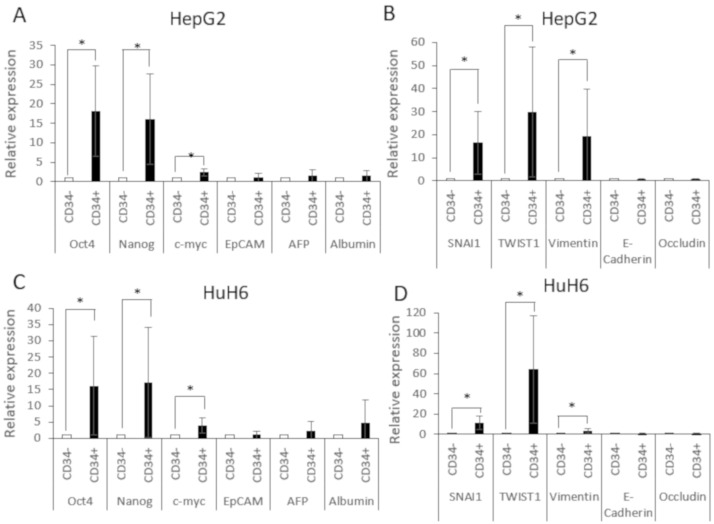
CD34^+^CD90^+^OV-6^+^csVimentin^+^ had increased expression of pluripotency and EMT factors. (**A**,**B**) Gene expression was analyzed in CD34 depleted (CD34^−^) and enriched (CD34^+^) fractions for Oct4, Nanog, c-myc, EpCAM, AFP, and albumin and for SNAI1, TWIST1, vimentin, E-cadherin and occludin in HepG2 cells and the same in HuH6 cells (**C**,**D**). The fold-change values of the CD34^−^ population were normalized to 1 and the values of CD34^+^ were calculated in relation to the respective values of CD34^−^. The columns represent the means with error bars depicting the standard deviation from the mean. The experiment was repeated at least 7 times. A two-tailed Wilcoxon ranked test was performed in order to calculate the significance of the data (* *p* < 0.05).

**Figure 3 cells-10-02598-f003:**
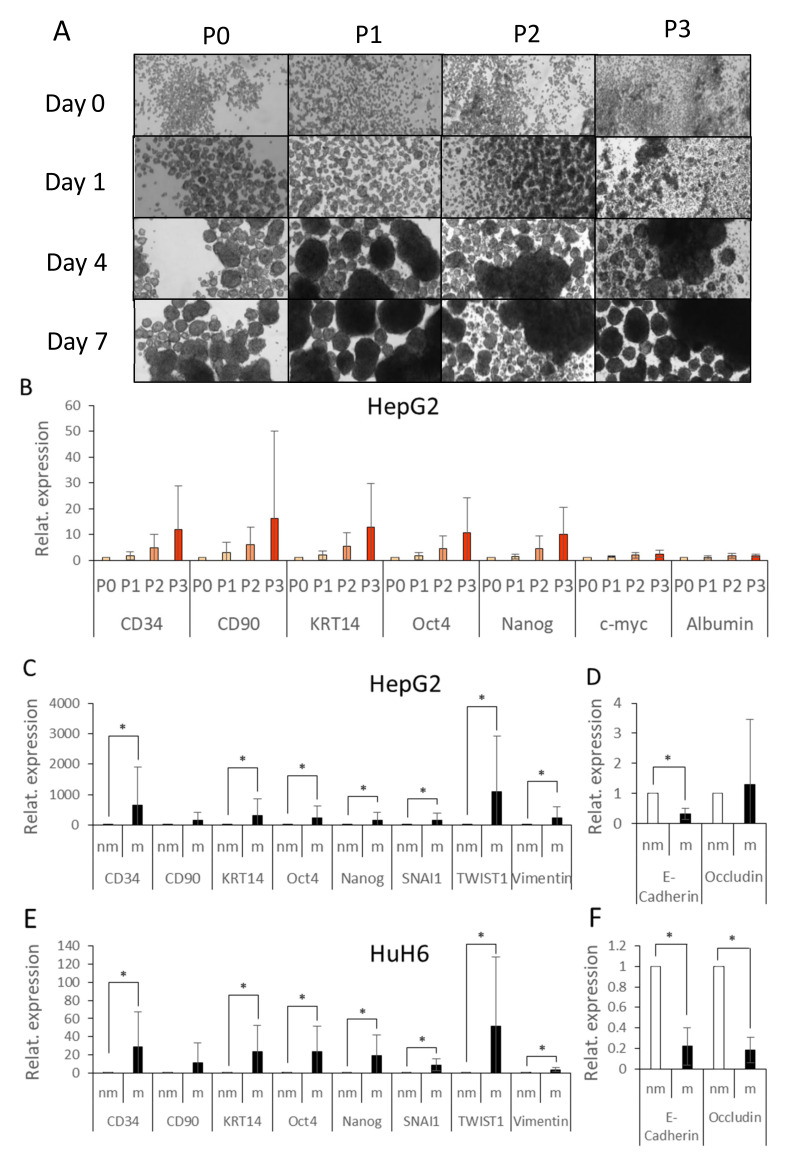
CD34^+^CD90^+^OV-6^+^csVimentin^+^ cells formed tumorspheres and migrated at a higher rate. (**A**) Tumorspheres of HepG2 cells were grown and passaged three times (P0–P3). (**B**) After 7 days of incubation, the gene expressions of CD34, CD90, KRT14 (one of the antigens of the OV-6 antibody), Oct4, Nanog, c-myc and albumin were measured by qPCR. The values of P0 were normalized to 1 and the fold changes of P1, P2 and P3 were calculated accordingly. The columns represent the mean with error bars depicting the standard deviation from the mean. The experiment was repeated 4 times. Dunn’s multiple comparisons test was performed in order to calculate the significance of the data. (**C**–**F**) HepG2 and HuH6 cells were seeded into a cell culture insert with a membrane containing media without FBS and placed into wells with media with FBS. After 24 h, the expressions of CD34, CD90, KRT14, Oct4, Nanog, SNAI1, Twist1, vimentin, E-cadherin and occludin were measured of the non-migrated (nm) and the migrated (m) cells using qPCR. The values of the non-migrated cells were normalized to 1 and the values of the migrated cells were calculated in relation to the non-migrated cells. The columns represent the mean with error bars depicting the standard deviation from the mean. The experiment was performed 4 times. A two-tailed Wilcoxon signed-rank test was performed in order to calculate the significance of the data (* *p* < 0.05).

**Figure 4 cells-10-02598-f004:**
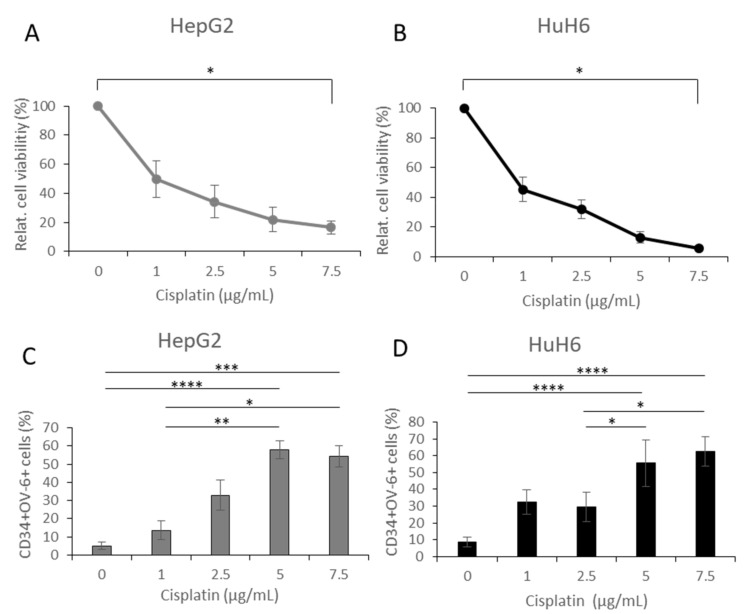
CD34+CD90+OV-6+csVimentin+ cells showed resistance toward Cisplatin, where the HepG2 and HuH6 cells were treated with 0, 1, 2.5, 5 and 7.5 µg/mL Cisplatin. (**A**,**B**) After 72 h, cell viability was measured in an MTT assay. The values are the means, which are presented with error bars depicting the standard deviation from the mean. The assay was performed 4 times. Dunn’s multiple comparisons test was performed in order to calculate the significance of the data (* *p* < 0.05). (**C**,**D**) Resisting cells were measured for CD34 expression and OV-6 binding using flow cytometry. The values are the means, which are presented with error bars depicting the standard deviation from the mean. The measurement was done 9 times. Dunn’s multiple comparisons test was performed in order to calculate the significance of the data (* *p* < 0.05, ** *p* = 0.001–0.01, *** *p* = 0.0001–0.001, **** *p* < 0.0001). (**E**–**R**) Gene expressions of Oct4, Nanog, SNAI1, TWIST1, vimentin, albumin and E-cadherin were measured for Cisplatin-treated HepG2 (**E**–**K**) and HuH6 cells (**L**–**R**). The fold-change values of the untreated cells were normalized to 1 and the values of the treated cells were calculated in relation to the respective values. The columns represent the means with error bars depicting the standard deviation from the mean (*n* = 4). Dunn’s multiple comparisons test was performed in order to calculate the significance of the data (* *p* < 0.05, ** *p* = 0.001–0.01).

**Figure 5 cells-10-02598-f005:**
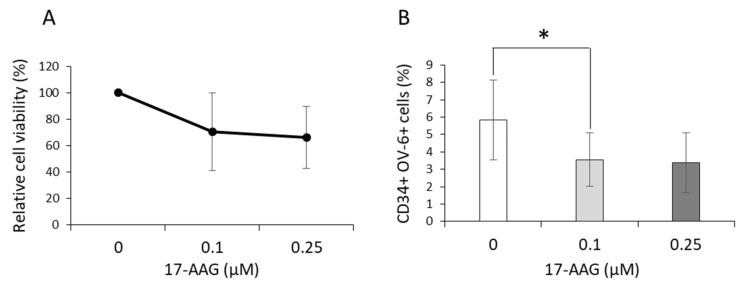
Treatment with 17-AAG-inhibited CD34+CD90+OV-6+csVimentin+ cells, where the HuH6 cells were treated with 0, 0.1 and 0.25 µM 17-AAG. (**A**) After 48 h, the cell viability was measured in an MTT assay. The presented values are the means with error bars depicting the standard deviation from the mean. Dunn’s multiple comparisons test was performed in order to calculate the significance of the data. (**B**) The cells were measured for CD34 expression and OV-6 binding using flow cytometry. The values represent the means with error bars depicting the standard deviation from the mean. The assay was performed 4 times. Dunn’s multiple comparisons test was performed in order to calculate the significance of the data (* *p* < 0.05). (**C**–**N**) Gene expressions of Oct4, Nanog, CD34, CD90, KRT14, SNAI1, TWIST1, EpCAM, albumin, c-myc, vimentin and E-cadherin were measured in qPCR experiments. The fold-change values of the untreated cells were normalized to 1 and the values of the treated cells were calculated in relation to the respective values. The columns represent the means with error bars depicting the standard deviation from the mean (*n* = 4). Dunn’s multiple comparisons test was performed in order to calculate the significance of the data (* *p* < 0.05). (**O**,**P**) Tumorsphere assays were performed with HuH6 cells, which were left untreated or treated with 0.1 µM 17-AAG. (**Q**) The average sphere size was calculated from 3 independent experiments. A two-tailed Wilcoxon signed-rank test was performed in order to calculate the significance of the data (* *p* < 0.05).

**Figure 6 cells-10-02598-f006:**
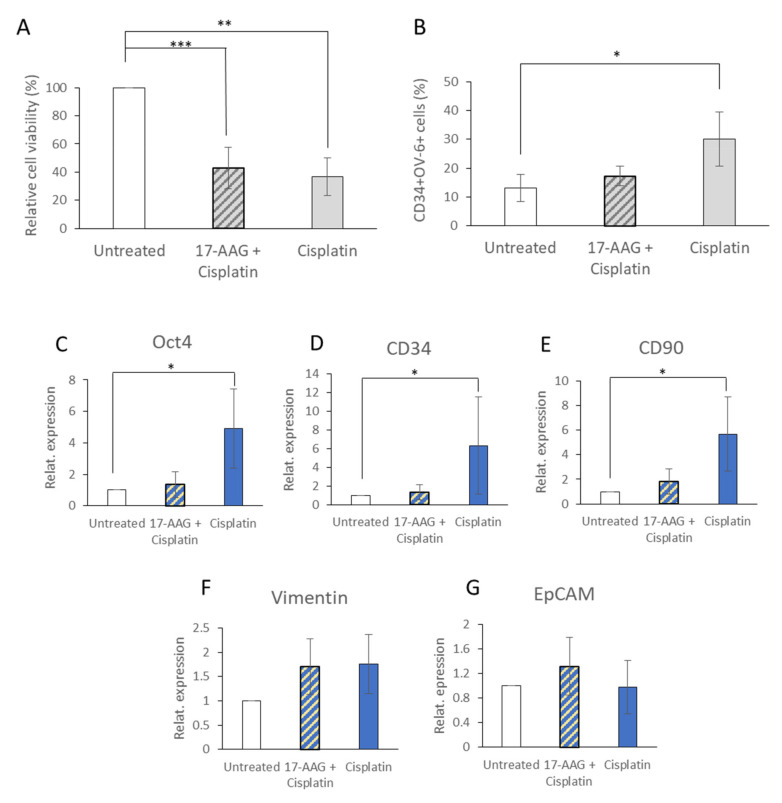
17-AAG even showed its effect on Cisplatin-treated cells. HuH6 cells were pretreated with 0.1 µM 17-AAG and subsequently with 2 µg/mL Cisplatin or treated with 2 µg/mL Cisplatin alone. (**A**) After 72 h of Cisplatin treatment, the cell viability was measured using MTT assays. The presented values are the means with error bars depicting the standard deviation from the mean. The assay was performed 4 times. Dunn’s multiple comparisons test was performed in order to calculate the significance of the data (* *p* < 0.05, ** *p* = 0.001–0.01, *** *p* = 0.0001–0.001). (**B**) Using flow cytometry, the expression of CD34 and OV-6 binding was measured in 5 independent experiments. The values represent the means and the error bars depict the standard deviation from the mean. Dunn’s multiple comparisons test was performed in order to calculate the significance of the data (* *p* < 0.05). (**C**–**G**) The expressions of Oct4, CD34, CD90, vimentin and EpCAM were evaluated using qPCR in 5 independent experiments. The presented values are the means and the error bars depict the standard deviation from the mean. Dunn’s multiple comparisons test was performed in order to calculate the significance of the data (* *p* < 0.05).

## Data Availability

The data presented in this study are available on request from the corresponding author.

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
