# Peer review of "Co-Expression of CD34, CD90, OV-6 and Cell-Surface Vimentin Defines Cancer Stem Cells of Hepatoblastoma, Which Are Affected by Hsp90 Inhibitor 17-AAG"

_cells, 2021, doi:10.3390/cells10102598_

Round 1

Reviewer 1 Report

In the manuscript entitled – “Co-expression of CD34, CD90, OV-6 and cell surface Vimentin 2 defines cancer stem cells of hepatoblastoma which are affected 3 by Hsp90 inhibitor 17-AAG”, the authors have identified a set of 4 co-expressing markers to identify cancer stem cells in hepatoblastoma cancer using two model cell lines. Further, they show enrichment of CSCs and gene expression markers upon cisplatin treatment. The identification of this CSC marker has very high translational significance, however, it needs to be validated further in multiple cancer cell lines or from primary tumor tissues. The significance of the 17-AAG drug is diluted given the fact that the effect is observed in one out of two cell lines tested.  The manuscript should be accepted for publication if the authors can address the following concerns:

  1. The authors should validate the 4-marker signature identified for Hepatoblastoma cancer in primary tumor samples or primary cell lines from tumors.
  2. A section on statistical analysis should be included in the methods section. The statistics are missing for Fig 3B- 3D, Fig4, Fig 5C - 5N, 5P, Fig 6, and Fig S4B.
  3. In Fig 4B, 5B, and 6B, the Y-axis shows CD34+ OV6+ Cells % whereas in the corresponding results the authors have mentioned checking the % of CD34+CD90+OV-6+csVimentin+ cells. Can the authors justify these differences?
  4. The authors should also include the background gating strategy used for identifying the CD34+CD90+OV-6+csVimentin+ subset by flow cytometry in Fig 4B, 5B, and 6B.
  5. Do others observe a synergetic effect on HuH6 or HepG2 cell survival when treated concurrently with cisplatin and 17-AAG or sequentially treating first by cisplatin to enrich CSCs and then by 17-AAG? Was the tumorsphere formation capability also altered?
  6. The mechanism of 17-AAG on CSC survival in HEpG2 cells needs to be investigated further. The authors can either try to justify why 17-AAG treatment does not lead to a decrease in the number of CSCs or include other hepatoblastoma cells to check for the correlation with the CSC population.

Author Response

First of all, we would like to thank the reviewers for their helpful comments and suggestions. According to their requests, we modified our manuscript and think that this improved the quality of our manuscript. We hope that the changes meet their requirements.

In general, both reviewers requested statistical analyses of the results. Therefore, we consulted a bio-statistician and performed the missing analyses. We also included a statistical part in the Material and methods section.

Please find below our responses to the specific comments and requests of the reviewer.

Sincerely,

Mieun Lee-Theilen

Responses to Reviewer 1

Query 1. The authors should validate the 4-marker signature identified for Hepatoblastoma cancer in primary tumor samples or primary cell lines from tumors.

Response 1. The reviewer made the eligible request to analyze further cell lines or primery tumor tissues. Here, we would like to explain the reasons why we presented our results only with two cell lines (HuH6 and HepG2). When we started our project, we attempted to get further hepatoblastoma cell lines and reached out to a research laboratory which established other hepatoblastoma cell lines. However, the research group did not reply to our contact attempts, so that we were not able to get these cell lines. Furthermore, we contacted Prof. La Quaglia (Memorial Hospital, New York) in whose laboratory a hepatoblastoma cell line was established in 1995. But, unfortunately, the cell line did not exist anymore and he could not send us an aliquot. We also attempted to establish a hepatoblastoma cell line by ourselves, when a hepatoblastoma patient was treated at our department in 2016. But also here, we were not lucky. As hepatoblastoma is a very rare tumor with an incidence of approx. 1.6 cases in 1 million children, the frequency of hepatoblastoma patients is very low. To avoid this lack of further cell lines, we would like to investigate other pediatric tumor entities in order to see if our markers apply to other CSCs as well and if 17-AAG also affects the CSC numbers of other tumor entities as seen in HuH6 cells.

Query 2. A section on statistical analysis should be included in the methods section. The statistics for Fig 3B-3D, Fig4, Fig 5C-5N, 5P, Fig 6 and Fig S4B.

Response 2. The statistical analysis of the figures has been performed and is mentioned in the according figure text as well as in the result text. An explanation of the statistical analyses was also included in the Material and methods section.

Query 3. In Fig 4B, 5B, and 6B, the Y-axis shows CD34+ OV6+ Cells % whereas in the corresponding results the authors have mentioned checking the % of CD34+CD90+OV-6+csVimentin+ cells. Can the authors justify these differences?

Response 3. The reason for this is based on the fact that we could observe the co-expression of all 4 markers on one subpopulation of HuH6 and HepG2 as presented in the Results part 3.1 shown in Figure 1C and 1D. This was also confirmed by enriching e.g. CD34+ cells and analyzing for the other markers by flow cytometry. There we could see that the other markers were enriched as well (shown in supplementary figure S1). Further on, we continued to only show the data of CD34+OV6+ cells representing this subpopulation positive for all 4 markers. We should have pointed this out more clearly and apologize for this confusion. We have added this information in the manuscript in various parts.

Query 4. The authors should also include the background gating strategy used for identifying the CD34+CD90+OV-6+csVimentin+ subset by flow cytometry in Fig 4B, 5B, and 6B

Response 4. The background gating strategy of the figures have been included into the supplementary figures S4-S7.

Query 5. Do others observe a synergetic effect on HuH6 or HepG2 cell survival when treated concurrently with Cisplatin and 17-AAG or sequentially treating first by Cisplatin to enrich CSC and then by 17-AAG? Was the tumor sphere formation capability also altered?

Response 5. Other studies addressed the effect of Cisplatin on CSCs in other tumor entities and could also observe an increase of CSCs using other CSC markers. We performed a Pubmed search using the search terms hepatoblastoma, cisplatin, 17-AAG, however, we did not find a publication which investigated the effect of Cisplatin and 17-AAG on CSCs of hepatoblastoma cells. Therefore, we believe that our study is the first address this question. We now mentioned this in the conclusion part. Furthermore, we only treated the cells first with 17-AAG and subsequently with Cisplatin, as we had the idea that 17-AAG might re-differentiate the CSCs and therefore become vulnerable to Cisplatin treatment. But it would be definitely interesting to see the effect of 17-AAG upon Cisplatin treatment.

We performed tumor sphere assays with double-treated cells. However, the formed spheres of the Cisplatin mono-treated cells did not look like the control spheres. Therefore, we assumed that Cisplatin treatment resulted in their resistance to this chemotherapeutic, but somehow affected their capability to grow to the same extent as under normal conditions.

Query 6. The mechanism of 17-AAG on CSC survival in HepG2 cells needs to be investigated further. The authors can either try to justify why 17-AAG treatment does not lead to a decrease in the number of CSCs or include other hepatoblastoma cells to check for the correlation with the CSC population.

Response 6. As the reviewer pointed out, the effect of 17-AAG on HuH6 was not observed in the HepG2 cells, which should be investigated further. Unfortunately, we are not able to investigate the effect in other Hepatoblastoma cell lines at the moment as explained earlier.

Like Cisplatin, 17-AAG in HepG2 cells seems to have a strong effect on “normal” cancer cells rather than the CSCs. In these cells, 17-AAG might induce the decrease of cell death inhibiting factors that results in cell death and leaving a supposedly increased number of CSCs.

A possible explanation for the missing effect on HepG2 CSCs could be that the key regulator or several factors which are responsible for cancer cell stemness are not stabilized by Hsp90, so that inhibition of Hsp90 does not affect these proteins and the stemness of the CSCs is not disturbed. It would be therefore also of great interest to identify the key regulators of CSC stemness in hepatoblastoma in the future. Furthermore, it could be still worth a try to investigate the impact of other Hsp90 inhibitors on HepG2 cells which have different mechanisms to affect the function of Hsp90.

We have modified this part in the discussion section.

Reviewer 2 Report

In this manuscript by Thielen et al., propose CD34, CD90, OV-6 and cell surface Vimentin (csVimentin) as markers to identify CSCs in hepatoblastoma cell lines. They show that the subset of CD34+CD90+OV-6+csVimentin+-co-expressing cells display pluripotency, self-renewal, increased expression of EMT markers and migration. Treatment with Cisplatin revealed chemoresistance of this sub- set within CSCs. When the cells were treated with the Hsp90-inhibitor 17- AAG, there was a significant reduction of the CSC subset. These are my major concerns:

In figure 3B,C,D there is very high variability within the replicates. These are not statistically significant. The authors need to either redo or explain the reason for this variability?

In figure 4 the authors compare HepG2 and Huh6 cell lines for cisplatin IC50, did they measure this for the spheroid model? It would be a good to see how IC50 is different between 2D and 3D models.

In figure 4C-P, I see no comparison between various cisplatin concentrations. The authors need to do statistical analysis.

What is rationale behind choosing 17-AAG?

In figure 5A,B there is very high variability within the replicates. These are not statistically significant. The authors need to either re do or explain the reason for this variability.

In figure 6A-G, I see no comparison between various treatment groups. The authors need to do statistical analysis.

Author Response

First of all, we would like to thank the reviewers for their helpful comments and suggestions. According to their requests, we modified our manuscript and think that this improved the quality of our manuscript. We hope that the changes meet their requirements now.

In general, both reviewers requested statistical analyses of the results. Therefore, we consulted a bio-statistician and performed the missing analyses. We also included a statistical part in the Material and methods section.

Please find below our responses to the specific comments and requests of the reviewer.

Sincerely,

Mieun Lee-Theilen

Responses to Reviewer 2

Query 1. In figure 3B,C,D, there is a very high variability within the replicates. These are not statistically significant. The authors need to either redo or explain the reason for this variability.

Response 1. The quantification of the genes of interest were normalized to an endogenous ACTB control and calculated by the 2(-DDCt) method. The high variability within the replicates in figure 3B can be explained as we experienced in one replicate much higher fold change results for all measured genes compared to the other replicates. Because of the n=4 only, we decided to leave this replicate in our results. We are not able to add more replicates due to the lack of time as this experiment takes four weeks. After consulting the bio-statistician, we performed a Dunn’s multiple comparisons test, which did not reveal any significance. Upon his suggestion, we left this anomalous replicate in the results.

Concerning Figure 3C and 3D, we repeated the experiments two more times and performed a non-parametric two-tailed Wilcoxon-signed rank test and added the results in the according result text and figure text.

Query 2. In figure 4, the authors compare the HepG2 and HuH6 cell lines for cisplatin IC50, did they measure this in the spheroid model? It would be a good to see how IC50 is different between 2D and 3D models.

Response 2. As the reviewer suggested, it would be interesting to compare the effect of Cisplatin in 2D with 3D models. We performed tumor sphere assays with Cisplatin treated cells. However, the formed spheres did not look like the control spheres. Therefore, we assumed that the cells were still viable, but somehow affected in their capability to grow to the same extent as under normal conditions. We did not check for cell proliferation, but microscopic analyses showed that the cells looked still viable for 11 days.

Query 3. In figure 4C-P, I see no comparison between various cisplatin concentrations. The authors need to do statistical analysis.

Response 3. A statistical analysis of the figures has now been performed and is mentioned in the according figure text and in the result text.

Query 4. What is the rationale behind choosing 17-AAG?

Response 4. As a chaperone protein, Hsp90 stabilizes various oncogenes and thus, was revealed to have an oncogenic function by itself. By inhibiting Hsp90, oncogenes and maybe also stemness factors could be destabilized and the consequence might be that the cancer cell or CSC becomes less tumorigenic or even dies.

So far, the hepatoblastoma therapy does not consider a targeted treatment against CSCs. One possible attack on CSC properties could be to target Hsp90 by utilizing an Hsp90 inhibitor such as 17-AAG which was found to affect the CSC population in osteosarcoma cells (Shu et al. Hsp90 inhibitor 17-AAG inhibits stem cell-like properties and chemoresistance in osteosarcoma cells via the Hedgehog signaling pathway. Onc Rep 2020, 44, 313-324). Our future plan is to identify such stemness factors which are degraded upon Hsp90 inhibition.

With this point, the reviewer made us aware that we haven’t explained enough our choice of 17-AAG. We modified this part in the discussion. 

Query 5. In figure 5A,B there is a very high variability within the replicates. These are not statistically significant. The authors need to either redo or explain the reason for this variability.

Response 5. The variability in the results seen in figure 5A could be explained by the idea that 17-AAG does not induce cell death, but rather modifies the CSCs. Therefore, we did not expect a statistically significant result. We now performed a Dunn’s multiple comparisons test, which is mentioned in the results and in the figure text.

Concerning figure 5A, we changed the statistical analysis towards a Dunn’s multiple comparisons test after consulting again the bio-statistician.

Query 6. In figure 6A-G, I see no comparison between various treatment groups. The authors need to do statistical analysis.

Response 6. A statistical analysis of the figures has now been performed and is mentioned in the according figure text as well as in the result text.

Round 2

Reviewer 1 Report

The authors have addressed the concerns raised in the first review process. The manuscript may be accepted in its current form.

Reviewer 2 Report

Please cite the following articles for 17-AAG:

  1. Sluder IT, Nitika, Knighton LE, Truman AW (2018) The Hsp70 co-chaperone Ydj1/HDJ2 regulates ribonucleotide reductase activity. PLoS Genet 14(11): e1007462. https://doi.org/10.1371/journal.pgen.1007462
  2. Aki Iwai, Dimitra Bourboulia, Mehdi Mollapour, Sandra Jensen-Taubman, Sunmin Lee, Alison C. Donnelly, Soichiro Yoshida, Naoto Miyajima, Shinji Tsutsumi, Armine K. Smith, David Sun, Xiaolin Wu, Brian S. Blagg, Jane B. Trepel, William G. Stetler-Stevenson & Len Neckers (2012) Combined inhibition of Wee1 and Hsp90 activates intrinsic apoptosis in cancer cells, Cell Cycle, 11:19, 3649-3655,